# Exercise Metabolome: Insights for Health and Performance

**DOI:** 10.3390/metabo13060694

**Published:** 2023-05-26

**Authors:** Aayami Jaguri, Asmaa A. Al Thani, Mohamed A. Elrayess

**Affiliations:** 1Weill Cornell Medicine-Qatar, Doha P.O. Box 24811, Qatar; 2Biomedical Research Center, Qatar University, Doha P.O. Box 2713, Qatar; 3QU Health, Qatar University, Doha P.O. Box 2713, Qatar

**Keywords:** exercise, metabolomics, metabolites, endurance exercise, resistance exercise, combined endurance–resistance exercise

## Abstract

Exercise has many benefits for physical and mental well-being. Metabolomics research has allowed scientists to study the impact of exercise on the body by analyzing metabolites released by tissues such as skeletal muscle, bone, and the liver. Endurance training increases mitochondrial content and oxidative enzymes, while resistance training increases muscle fiber and glycolytic enzymes. Acute endurance exercise affects amino acid metabolism, fat metabolism, cellular energy metabolism, and cofactor and vitamin metabolism. Subacute endurance exercise alters amino acid metabolism, lipid metabolism, and nucleotide metabolism. Chronic endurance exercise improves lipid metabolism and changes amino acid metabolism. Acute resistance exercise changes several metabolic pathways, including anaerobic processes and muscular strength. Chronic resistance exercise affects metabolic pathways, resulting in skeletal muscle adaptations. Combined endurance–resistance exercise alters lipid metabolism, carbohydrate metabolism, and amino acid metabolism, increasing anaerobic metabolic capacity and fatigue resistance. Studying exercise-induced metabolites is a growing field, and further research can uncover the underlying metabolic mechanisms and help tailor exercise programs for optimal health and performance.

## 1. Introduction

Exercise benefits every part of the human body and plays an important role in preventing chronic diseases. The effects of exercise are mediated by a complex process involving the activation of integrated body systems at the molecular and cellular levels. The growing use of metabolomics technologies in this field has made it possible to comprehensively study the exercise metabolome and understand the many benefits of exercise on the mental and physical wellbeing of humans. Tissues such as skeletal muscle, bone, and the liver were found to release metabolites into our blood [1,2] and are involved in pathways for respiration, lipolysis, and metabolic stress, helping to improve cardiovascular fitness and maintain muscle mass. In addition, metabolomics approaches helped identify the underlying mechanisms, pathways, and biomarkers involved in diseases and aging processes. For example, many exercise-induced metabolites were found to play key roles in the pathways involved in oxidative stress regulation, neurotrophic signaling, and autophagy, which were documented to positively influence a range of ailments including muscular aging, neurodegeneration, and heart disease [3,4,5,6]. Metabolomics is also a promising tool for the early diagnosis of diseases that cannot be treated, such as Alzheimer’s, helping to identify biomarkers and aiding in assessing therapeutic efficacy [7].

Moreover, many of these exercise-induced changes can lead to long-term metabolic adaptations, as documented in studies on elite and professional athletes [1,8,9,10] as well as chronic exercise interventions [11,12,13]. In athletes, for example, years of training coupled with a unique genetic makeup result in significant metabolic adaptation specific to their type of sport: endurance training tends to increase mitochondrial content and the activity of oxidative enzymes [14], and resistance training is associated with increases in muscle fiber and increased glycolytic enzymes in skeletal muscle [15]. As this is an emerging field, there are potentially many underlying metabolic mechanisms that have yet to be uncovered; further elucidation of these processes could help to identify potential therapeutic targets for athletes and the general population as well as provide the crucial information needed for the optimal balance between exercise training and recovery, allowing exercise programs to be finely tailored to individuals and athletes.

## 2. Metabolomics

Metabolomics is a combined set of strategies to identify and quantify cellular metabolites using advanced analytical tools. This is typically achieved through the use of liquid or gas chromatography, which allows for the detection of individual metabolites through their specific mass-to-charge ratio (*m*/*z*) and their fragmentation in a mass spectrometer [16]. By matching detected metabolites against databases of known metabolites, it is possible to identify the specific metabolites altered by exercise in a biological sample [17]. This allows researchers to characterize metabolic differences between groups and individuals, helping them to understand the biological mechanisms underlying key differences. The metabolome can indicate past and present exposure to exercise [8,18,19,20], reflecting the physiological adaptations of an individual and even predicting their future responses. The increasing body of literature involving metabolomics indicates its growing role in the field of sports medicine.

### 2.1. Diagnostic and Therapeutic Use

Metabolomics is widely regarded as a promising tool for diagnostics, which is used to determine the significant biological pathways playing a role in disease development. Blood-based metabolomics, for example, is ideal for the early detection of a wide variety of diseases, such as breast cancer, leukemia, Alzheimer’s, and heart failure [21]. In addition, metabolomics can be used to determine potential targets for therapeutic intervention by identifying pathological metabolic changes. For example, the examination of the metabolites up-regulated by the progression of prostate cancer points to the role of sarcosine in promoting cancer cell invasiveness, helping to identify the enzyme responsible for sarcosine production as a potential therapeutic target [22]. Metabolomics identifies biomarkers for the diagnosis, therapy, and monitoring of disease—making further classification of disease possible and enabling treatment options to be more specifically tailored to patients.

### 2.2. How Does Exercise Affect Metabolism?

The metabolome, which comprises all metabolites within an organism, is highly susceptible to external influences such as exercise. The literature to date overwhelmingly concludes that different degrees of physical exercise are responsible for quantifiable changes in the human metabolome, as measured in biological samples. These metabolic alterations are most commonly characterized by changes in lipid metabolism, amino acid metabolism, energy metabolism, carbohydrate metabolism, and nucleotide metabolism, among others [2,8,10,18]. Across the literature, findings widely reflect the known positive impacts and beneficial physiological adaptations linked to physical exercise. All reported metabolic pathways are inextricably linked: they play interconnected roles in aerobic and anaerobic respiration, fatty acid oxidation, branched-chain amino acid catabolism, and oxidative stress. While there is some debate in the literature about which components of exercise lead to the most pronounced changes, the research overwhelmingly supports a multifactorial dose relationship between exercise and the human metabolome, with factors such as the duration, intensity, and type of activity causing significant changes in the metabolic response.

## 3. Types of Exercise

Types of exercise with a high cardiorespiratory component can be classified as endurance exercise, such as running, cycling, and soccer [23]. Resistance exercise, on the other hand, typically has a low cardiorespiratory component. It is, instead, designed to improve muscular strength, through the use of weights, for example. Sports such as gymnastics, martial arts, and climbing also demonstrate a high resistance component. Moreover, endurance and resistance components are often combined, for example, in exercise interventions incorporating running alongside weight training. Many sports also have both a significant endurance and resistance component, for example, sprinting, boxing, and rugby.

Diverse sports training interventions are classified in this review according to their specific endurance and resistance demands. Sports with a high endurance demand (>70% Max O_2_) and a low resistance demand (<20% Maximal Voluntary Contraction; MVC) are included in the endurance section; those with a high resistance demand (>50% MVC) and low endurance demand (<40% Max O_2_) are in the resistance section, while those with similar endurance and resistance demands are placed in the “combined endurance–resistance” section.

Furthermore, this review also classifies exercise based on its duration. Acute exercise is typically considered to be a short, singular bout of exercise [24]. Meanwhile, chronic exercise is commonly described as exercise carried out over a duration of 6 weeks or greater [25]. Subacute exercise is categorized in this review as falling between acute and chronic—exercise that is more than a single bout but lasts less than 6 weeks.

### 3.1. Effects of Exercise Modality

Endurance exercise (i.e., exercise involving cardiorespiratory aerobic components such as jogging, treadmill running, and cycling) was shown to positively influence cardiovascular fitness and mitochondrial biogenesis [14]. It promotes improvements across the key parameters of aerobic fitness, including maximal oxygen uptake (VO_2_ max), the lactate/ventilatory threshold, oxygen uptake kinetics, and exercise economy. This allows individuals to exercise for longer periods at the same intensity or maintain a higher intensity for the same duration.

Important metabolomic adaptations linked to endurance exercise help to promote aerobic processes and a highly oxidative metabolism as well as a decreased glycolytic metabolism [8]. Energy is produced through oxidative phosphorylation during the prolonged aerobic exercise involved in endurance training [26]. The metabolic pathways involved in the oxidative system are, therefore, activated, leading to greater beta-oxidation of fats. Endurance exercise is also distinguished by greater fuel diversity during exercise, which is indicated by the metabolites of the carbohydrate metabolism, amino acid metabolism, and fatty acid metabolism (in contrast to resistance exercise, which solely requires carbohydrate substrates) [27].

On the other hand, resistance training involves exercises by a muscle or muscle group against external resistance, typically comprising of high-load, low-repetition muscle contractions (i.e., weight training, plyometric training, and machine-based training). It is known to induce muscle hypertrophy in addition to improving muscular strength and metabolic health [28].

Resistance training is linked to the metabolic changes that facilitate improved anaerobic capacity, muscular fitness, and glycolytic metabolism [15]. Among the most prominent metabolic adaptations induced by resistance training is the elevation of protein synthesis and amino acid depletion, which are required to increase muscle mass [8,18,29]. In addition, the pathways involved in nucleotide synthesis—needed to produce RNA, DNA, and phospholipids for cell membranes—are activated [2,27,30]. Both the rates of ATP hydrolysis and nucleotide turnover are shown to be increased during acute resistance exercise [18]. As the anaerobic processes stimulated by resistance exercise lead to exhausted ATP-creatine phosphate (ATP-CP) stores, muscles resort to ATP regeneration through the breakdown of glycogen via the glycolytic pathway [26]. This is reflected by the increased glycolytic metabolism and lactate accumulation in the metabolome following acute resistance exercise [2,18,30] as well as by the metabolic adaptations promoting increased glycolytic capacity in response to chronic resistance exercise [8,18].

When properly combined, concurrent endurance and resistance training can provide benefits to both aerobic capacity and muscular strength. Some studies proposed an interference effect associated with this combined modality of exercise, in which the metabolomic adaptation to endurance and resistance training is diminished in comparison to separately training either exercise modality [31]. For sports requiring both high endurance and strength, this creates the challenge of maximizing the adaptation to both endurance and resistance training, while minimizing the conflict between the two. Interestingly, some studies also provided evidence contrary to the interference phenomenon, showing that endurance exercise combined with resistance can improve muscle growth, provided certain conditions are followed and enough rest is given between bouts [32,33].

### 3.2. Effects of Exercise Duration

The literature to date overwhelmingly supports both an acute metabolomic response to a bout of exercise as well as a chronic metabolic adaptation to exercise performed over a prolonged period of time. Acute exercise is characterized in this review as a short burst of exercise, performed in a singular bout [24], while chronic exercise is categorized as exercise that occurs over a longer duration of 6 weeks or more [24]. Lastly, subacute exercise is categorized in this review as exercise that falls between acute and chronic—more than a single bout but that lasts less than 6 weeks. Notably, the length and intensity of the physical activity performed were also shown to have an effect on the extent of the metabolic changes and the duration for which they persist [34]. The metabolic effects of long-term, sustained exercise are critical to understand, as they carry important implications for exercise interventions related to sports training, weight loss, fitness, and general public health.

The literature overwhelmingly supports the existence of an acute metabolomic response to exercise, often characterized by widespread changes involving the pathways that are central to cardiometabolic health and cardiovascular disease. Key examples of this acute metabolomic response include changes in amino acid metabolism, lipid metabolism, and increased beta-oxidation of fats following acute endurance exercise [35,36,37] as well as the alteration of the pathways involved in glycolysis, the tricarboxylic acid (TCA) cycle, and nucleotide metabolism following acute resistance exercise [2,27,30].

Several studies also support the existence of a chronic metabolomic adaptation in response to exercise performed over a prolonged period of time. These include metabolic changes that enrich the pathways involved in the oxidative systems in endurance athletes, while reducing glycolytic capacity [8]. In contrast, in resistance athletes, chronic metabolomic adaptations are often characterized by metabolic changes enhancing anaerobic processes and muscle strength, primarily in the pathways facilitating glycolysis, protein synthesis, amino acid depletion, and nucleotide turnover [8,18].

Chronic metabolomic adaptations are perhaps most prominently reflected in a subset of the literature targeting elite athletes, for whom metabolic adaptations are likely to be the most pronounced due to their extensive, high-intensity training over a period of several years. The literature points to an altered metabolome in elite athletes, signaling underlying metabolic adaptations that enhance fuel substrate utilization, the beta-oxidation of fatty acids, oxidative stress, steroid biosynthesis, and protein synthesis, among other adaptive processes [8,9,10,38].

### 3.3. Effects of Overexercising

Exercise should be performed within healthy limits; too much exercise can be to the detriment of the individual. The clinical and endocrinological symptoms of overexercising can include mood changes, fatigue, muscle pain, and disruptions in sleep. In addition, overexercising is linked to delayed-onset muscle soreness (DOMS), which typically occurs 24 h after the overexercise episode. This muscle soreness can persist for days and may even progress to rhabdomyolysis, causing permanent muscle damage or even death [39]. Repetitive muscle contraction can lead to microtraumas in the tissue, leading to acute inflammation that, over time, can progress to chronic inflammation. Cortisol depletion was also implicated in overexercising and could lead to fatigue disorders as a result [40]. In addition, a decrease in BCAA levels following a heavy bout of exercise was associated with overtraining symptoms of fatigue. This may increase the uptake of tryptophan in the brain, thereby increasing 5-HT synthesis and inducing fatigue [40]. Overall, these findings highlight the healthy limits of exercise: with adequate rest and recovery, exercise can provide tremendous benefits.

The overall findings for each type of exercise based on its type (endurance/resistance) and duration (acute/subacute/chronic) are discussed in Section 4, Section 5 and Section 6, in addition to the metabolomic changes distinguishing elite athletes (assumed to have the most pronounced metabolic adaptations to their respective disciplines). The included studies are also described in Appendix A.

## 4. Metabolic Signature of Endurance Exercise

### 4.1. Acute Endurance Exercise

Acute exercise is defined as a short burst of exercise, performed in a single bout [24]. Acute endurance exercise (i.e., a bout of long-distance running or using a bicycle ergometer) is linked to the increased consumption of energy and the decreased efficiency of ATP synthesis, with the most pronounced metabolic effects occurring at a high intensity (≥85% of VO_2_ peak) [41].

Several metabolomics studies showed that endurance exercise at high-intensity and acute duration induces changes in amino acid metabolism (Appendix A). The pathways found to be enriched by this form of exercise include alanine and aspartate metabolism, glutamate metabolism, arginine biosynthesis, arginine and proline metabolism, threonine and 2-oxobutanoate degradation, and tryptophan degradation. The activation of the glutamate metabolism pathway after exercise was noted in a number of studies [20,35,36,42]—including Zhao et al., which reported a significant upregulation—and is indicative of lower cardiometabolic risk. Alteration of the urea cycle, which is responsible for ammonia detoxification, was also noted in response to inflammation induced by exercise [27,42,43]. Moreover, increased tryptophan degradation and upregulation of the kynurenine/tryptophan metabolic pathway was reported following acute high-intensity endurance exercise interventions [44,45]. While one study—Schenk et al.—did not find a statistically significant alteration in kynurenine and tryptophan metabolites, it noted the resting control group showed lower levels of tryptophan degradation three weeks after the intervention than the intervention group [46]. There is an overall decrease in the circulating levels of the metabolites associated with insulin resistance and cardiovascular risk (i.e., glutamate and BCAAs) and an increase in those associated with higher inflammation (i.e., arginine and kynurenine) [19,34,35,43,44]). Pinto et al., noted that amino acid utilization differed between the sexes.

Additionally, accelerated fat metabolism was observed in a number of studies, with significant changes across all forms: long-chain, medium-chain, branched-chain, am-ino-FAs (amino-fatty acids), carnitines, ketones, lipid mediators, membrane lipids, and fatty acid esters of hydroxy fatty acids (FAHFAs) [2,4,20,27,35,36,42,45,47,48,49]. Studies also highlighted ketone body accumulation following exercise, suggesting a switch from glycolytic metabolism to ketolytic metabolism during acute endurance exercise [2,27,35,45,48]. The primary ketone bodies acetoacetate and β-hydroxybutyrate increased in response to acute endurance exercise [27,35,45] but not to resistance exercise [27]. A possible reason for this difference could be the higher caloric expenditure needed for acute endurance exercise, which leads to an energy demand exceeding intracellular supplies, resulting in ketone bodies being delivered from hepatocytes. Alterations in steroid metabolism were also observed [20,27,42]. Zhao et al., found steroid metabolism to downregulate after exercise and upregulate after recovery, while the metabolites related to lipid transport and metabolism were significantly perturbed during exercise and recovery. Pinto et al. noted a decrease in cortisol following acute endurance exercise as well as an increase in testosterone, dehydroepiandrosterone (DHEA), estrone, and 17-OH-progesterone. Testosterone was shown to increase muscle mass and strength [50], while estrogen decreased bone resorption and muscle damage [51]. Nemkov et al., additionally observed an overall decrease in RBC lipid levels, indicating the activation of the membrane lipid remodeling pathways [52]. Interestingly, Varga et al., noted a blunted metabolomic response in the lipid profiles of female athletes with Relative Energy Deficiency in Sport (RED-S) syndrome, which is often caused by overtraining and inadequate nutrition [47]. Athletes with RED-S showed a significantly smaller increase in phosphatidylethanolamines levels and a markedly different trajectory of triglyceride levels than athletes with sufficient energy availability, indicating that overtraining likely hinders the adaptive metabolic response to exercise interventions.

Changes in the metabolism of cofactors and vitamins were also documented with the pathways of CoA biosynthesis, and pantothenate and CoA metabolism were activated [4,20]. Tetsuyuki et al., noted an elevated concentration of urobilinogen following the completion of a marathon, indicating accelerated hemoglobin and porphyrin metabolism; this may be attributable to increased exercise-induced hemolysis. In addition, acute high-endurance exercise was linked to major changes in the metabolites related to cellular energy metabolism, such as the TCA that was cycle-marked by alterations of succinate, fumarate, malate, and 2-ketoglutarate-and glycolysis [2,20,27,36,43,53]. Among the most pronounced of these changes is the elevation of succinate, which plays the role of a fuel in thermogenic adipocytes to promote energy expenditure. A number of studies found carbohydrate metabolism to be affected [2,20,27,36,43,53], with significant increases detected in lactate and alpha-ketoglutarate [2,53], indicating the upregulation of glycolysis metabolism. Purine metabolism was also significantly altered [2,20,43]: Morville et al. found the purine metabolism pathway to be upregulated, with several of the purine metabolites remaining elevated 3 h post-exercise.

### 4.2. Subacute Endurance Exercise

Subacute exercises falls between acute and chronic—lasting longer than acute (a short, single bout of exercise) but shorter than chronic (duration of 6 weeks or more) [25]. Examples of subacute endurance exercise include 1–6 week long exercise interventions involving long-distance running or the use of a bicycle ergometer (e.g., 10-day HIIT on a bicycle ergometer, [54]).

Several studies reported the metabolic changes associated with subacute endurance exercise (Appendix A). Changes in amino acid metabolism were observed following subacute endurance exercise [44,54,55,56]. An increase in plasma urea was noted [56], in addition to alterations of cadaverine, lysine, and N6-acetyllysine, which are involved in lysine metabolism [55]. Moreover, pathways linked to leucine metabolism, isoleucine metabolism, and valine metabolism as well as taurine metabolism were noted to be altered: Kistner et al. reported increases in methylsuccinate and taurine concentrations from baseline to 4 days after recovery following the exercise intervention [54]. Tryptophan metabolism was also affected: Joisten et al. reported activation of the kynurenine pathway marked by a decrease in tryptophan levels post-exercise, immediately after high-intensity exercise and 3 h after moderate-intensity exercise [44].

With respect to lipid metabolism, pathways linked to primary bile acid metabolism were enriched, stimulating increases in acyl-carnitines, FAs, diacylglycerols, and primary bile acids [57]. The concentrations of acyl-carnitines, TCA cycle metabolites, and lipid metabolites were elevated, indicating heightened lipid mobilization and mitochondrial substrate shuttling following subacute endurance exercise. Altered nucleotide metabolism was also observed: Kistner et al. found a significant decrease in hypoxanthine concentration after exercise, which is indicative of a training-induced adaptation of purine metabolism [54].

### 4.3. Chronic Endurance Exercise

Chronic exercise typically occurs over a longer duration of 6 weeks or more [25] and is associated with favorable metabolic and cardioprotective adaptations. Examples of this kind of exercise can include endurance-based exercise programs performed over an extended time (e.g., an 8-week training program on a bicycle ergometer [12]). In addition, long training periods of sports with a high endurance demand (and a low resistance demand)—such as soccer, tennis, and cross-country skiing—also fall under this classification. Notably, elite athletes—athletes competing at the professional, national, or international level—in such endurance-based sports comprise an important subset of the literature on chronic exercise, as it is assumed their metabolic adaptations are the most pronounced. In particular, elite endurance athletes have distinct metabolic adaptations facilitating aerobic processes, promoting a highly oxidative metabolism, and decreasing glycolytic metabolism [8].

Many studies investigated the metabolic signature associated with chronic endurance exercise (Appendix A). Improvements in lipid metabolism were documented [8,9,10,11,12,13,19,58,59], with significant variations observed in the concentrations of 3-hydroxybutyric acid and acetoacetate (involved in ketolytic metabolism), and trimethylamine-N-oxide (utilized in phospholipid metabolism). Long-term metabolomic adaptations to chronic endurance exercise indicate an overall healthier lipid profile. This is characterized by elevated concentrations of total High Density Lipoprotein (HDL) cholesterol and the subfraction High Density Lipoprotein 2 (HDL2), a higher average HDL particle size, a higher cholesterol to phospholipid ratio, and elevated concentrations of the specific phospholipases linked to cardiometabolic health [8,9]. These findings suggest an improved efficiency of cholesterol excretion as an adaptation to chronic endurance exercise, through one of two mechanisms—transintestinal cholesterol excretion or bile salts. In addition, the pathways involved in fatty acid (acyl choline) metabolism, medium chain fatty acid metabolism, and primary bile acid metabolism were also noted to be enriched [19], with elevations of the primary bile acid metabolites cholate and chenodeoxycholate. In addition to serving as digestive surfactants, bile acids also function as important signaling molecules for lipid metabolism and energy metabolism.

Subjects trained in chronic endurance exercise were also noted to have lower levels of specific phosphatidylcholines (PCs), perhaps as these are required as ligands for signaling processes in adaptation [8,59]. It was also posited that endurance-trained subjects use long-chain fatty acids differently than non-endurance trained subjects: they were observed to show greater increases of medium/long-chain acylcarnitines, which are involved in the beta-oxidation of long-chain fatty acids, than other groups [8]. Adaptations of steroid metabolism were also noted—Tarkhan et al. demonstrated steroids from the androgen, pregnenolone, and progestin subpathways to be decreased in elite female endurance athletes, while corticosteroids were increased [10]. The present literature on androgenic steroids shows them to enhance sports performance by impacting muscle tissue, bone mass, and erythropoietin and behavioral patterns [60]. A potential explanation for upregulated cortisol pathways could be the adaptations affecting carbohydrate utilization in elite female endurance athletes.

Moreover, a number of studies found chronic endurance exercise to induce changes in amino acid metabolism [11,12,13,61,62]. Pintus et al., noted the enrichment of the pathways involved in creatine metabolism, with significant changes in the concentrations of guanidoacetic acid and creatine observed following exercise [13]. Additionally, changes in energy metabolism were noted [11,13,61,62]. Variations in thiaminetriphosphate concentrations—an ester of the vitamin thiamine playing a role in glucose regulation—were observed following chronic endurance exercise.

Interestingly, Serra et al., noted the activation of heparan-, chondroitin-, and keratan-sulfate degradation (involved in connective tissue metabolism) and linoleate metabolism (linked to lipid signaling) in chronic stroke survivors [63]. While these pathways do not match those previously recorded in non-stroke populations, these results indicate that the metabolomic response to exercise may differ between study populations.

## 5. Metabolic Signature of Resistance Exercise

### 5.1. Acute Resistance Exercise

Acute resistance exercise typically involves high-load, low-repetition muscle contractions [64] performed in a single bout. Examples could include weight training, plyometric training, or machine-based training, comprising upper and lower body exercises such as squats, jumping, sprinting with a load, push-ups, and pull-ups [65]. This modality of training is known to facilitate metabolic changes that promote anaerobic processes and increase muscular strength.

Many studies indicated the metabolic signature associated with acute resistance exercise (Appendix A). Following a bout of acute resistance exercise, significant alterations of energy metabolism and carbohydrate metabolism—including the pathways involved in glycolysis and the TCA cycle—were noted. This was observed through corresponding increases in succinate, lactate, pyruvate, malate, fumarate, 2-oxoglutarate, and α-ketoglutarate [2,18,27,30,66,67,68]. Morville et al., noted the activation of glycolysis and the TCA cycle pathways to be substantially more pronounced for resistance exercise than for endurance training, indicative of the demand for higher ATP turnover [2]. Pechlivanis et al., noted there were no significant changes in ketone bodies, acetoacetate, or 3-hydroxybutyrate, indicating a lack of effect on fatty acid degradation—as expected, since a shortage of carbohydrates and the corresponding need for fatty acid metabolism does not typically occur during brief resistance exercise [30]. In addition, Gehlert et al., observed a significant increase in bile acids, which act as the signaling molecules to regulate energy metabolism in addition to enabling the absorption of lipids [18].

In addition, nucleotide metabolism was affected—studies noted the activation of the purine degradation pathways with increased concentrations of hypoxanthine, xanthine, xanthosine, and inosine [2,18,27,30,67]. Significant degradation of nicotinamide adenine dinucleotide (NAD) was also noted by Gehlert et al., reflecting a need for increased NAD turnover and ATP production during acute resistance exercise [18]. Nucleotide metabolism is known to be a key component of the adaptive response to high-intensity resistance exercise across all ages, as it is essential for ATP resynthesis during exercise [69].

Lastly, changes in amino acid metabolism also took place. Studies noted alterations in the breakdown products of Branched-Chain Amino Acids (BCAAs), histidine metabolism, tryptophan metabolism, lysine metabolism, alanine and aspartate metabolism, and arginine and proline metabolism [2,18,27,30,67].

### 5.2. Chronic Resistance Exercise

Chronic resistance exercise involves resistance exercise performed over a period of 6 weeks to several months. It can include strength training sessions regularly performed for an extended duration as well as sports with a high resistance component and a low endurance component, such as gymnastics, weight lifting, body building, climbing, and the martial arts. Elite athletes competing in these sports for several years display some of the most pronounced adaptations to chronic resistance exercise in their metabolome.

Several studies highlighted the metabolic changes associated with chronic resistance exercise (Appendix A). Changes in amino acid metabolism were noted. Amino acids (i.e., alanine, lysine, and methionine) were observed to be lowered after the chronic resistance exercise intervention. Among the most significantly altered metabolic pathways were those involved in glutamate metabolism, alanine and aspartate metabolism, and arginine and proline metabolism [18,29,70]. Gehlert et al. also observed lowered levels of N-acetylated ketogenic amino acids and elevated levels of beta-citrylglutamate, which was linked to improved skeletal muscle growth [18].

Energy metabolism was also shown to be significantly affected. Gehlert et al. noted decreased levels of fructose 1-6-bisphosphate, which is involved in glycolysis, and acetylphosphate, which is involved in ATP resysnthesis [18]. This may indicate a heightened oxidative capacity of skeletal muscle in conjunction with a reduced glycolytic capacity in response to chronic resistance exercise.

Moreover, studies detected several changes in lipid metabolism. Ketone body metabolism was shown to be affected in a study by Gehlert et al., in which a significant decrease in BHBA (3-hydroxybutyrate) was observed [18]. This may indicate an increased capacity to oxidize the BHBA induced by chronic resistance training or a chronic fall in the metabolite’s baseline levels. Additionally, a significant elevation of lysophosphatidylcholine concentration was noted after the chronic resistance exercise intervention in Shen et al. [29]. Interestingly, Ahmeti et al. noted resistance training induced a comparable lipid profile to that caused by endurance training [58]. Although endurance training is expected to promote greater improvements to the lipid profile than resistance training, a number of studies nevertheless documented chronic resistance training as inducing significant improvements to cardiovascular risk factors [71,72].

Moreover, Mieszowski et al., found the chronic-resistance-based training of elite gymnasts to induce an adaptive response affecting metabolic markers of bone formation and resorption [1]. Concentrations of the bone resorption metabolite CTX (C-terminal telopeptide of type 1 collagen) were found to be significantly elevated in the non-athlete controls post-training but not for the athletes, suggesting a protective adaptation in the athletes’ skeletal muscle during the resorption state.

## 6. Metabolic Signature of Combined Endurance and Resistance Exercise

Combined endurance–resistance exercise consists of both significant aerobic and anaerobic components. For instance, interventions involve both endurance and resistance training, such as Lee et al., which had two whole-body resistance training sessions and two bicycle interval sessions each week for 12 weeks [73]. This type of exercise also includes sports with both significant endurance demand and resistance demand, such as swimming and basketball. It was suggested that combined endurance–resistance exercise carries greater benefits for overall health—including cardioprotective markers and muscular fitness—than either exercise modality alone [74].

### 6.1. Subacute Combined Endurance and Resistance Exercise

A study on subacute combined endurance–resistance training identified changed metabolites that were primarily involved in energy metabolism, lipid metabolism, and amino acid metabolism [75]. Increases in several amino acids, including alanine, valine, isoleucine, leucine, glycine, serine, threonine, and ornithine, were noted in the post-training group when compared with the pre-training group. Several metabolic pathways were described as disturbed, including glutathione metabolism, alanine, aspartate and glutamate metabolism, glyoxylate and dicarboxylate metabolism, arginine biosynthesis, glycine, serine and threonine metabolism, linoleic acid metabolism, arginine and proline metabolism, D-glutamine and D-glutamate metabolism, the citrate cycle, and aminoacyltRNA biosynthesis.

In addition, Loureiro et al., noted a marked increase in lactate over each week of subacute combined training, indicating the enrichment of the pathways involved in glycolysis, gluconeogenesis, and pyruvate metabolism [76]. Alterations in amino acid metabolism were also noted, with significant changes in the concentrations of creatine (involved in creatine metabolism) and sarcosine (involved in glycine metabolism, serine metabolism, and threonine metabolism).

The involvement of purine metabolism was also noted following subacute combined exercise [77]. Wang et al., highlighted significant changes in the related metabolites, L-glutamine, AMP (adenosine monophosphate), and hypoxanthine. Of these, hypoxanthine was found to be upregulated 6.73 times that of the pre-training levels, indicating enhanced ATP metabolism as the exercise progresses. Although post-competition Hx remained upregulated 4 weeks after training, the degree of upregulation compared to pre-training was reduced, indicating an improvement in athletes’ anaerobic metabolic capacity and fatigue resistance after 4 weeks of training. Furthermore, Wang et al. noted the involvement of nicotinate and nicotinamide metabolism—the relative levels of two of the metabolites involved (NAM and N1-methylnicotinamide) decreased significantly, while 2-pyridone increased significantly after water polo games. Studies reporting metabolic changes associated with combined endurance–resistance training are summarized in Appendix A.

### 6.2. Chronic Combined Endurance and Resistance Exercise

During chronic combined endurance–resistance exercise, several studies showed alterations of lipid metabolism [37,38,73,78,79,80,81,82]. These included alterations of certain phospholipids, diacylglycerols, ceramides, acylcarnitines, cardiolipins, cortisols, ketone bodies, HDLs, and unsaturated fatty acids. In one study, chronic combined exercise was shown to increase the ratio of phosphatidylcholines (PC) to phosphatidylethanolamines, driven by an increase in PC, which may reflect the increased demand and synthesis of PCs [80]. Furthermore, combined exercise training also increased the skeletal muscle acylcarnitines linked to fatty acid mobilization and the availability of TCA cycle intermediates [80,82], decreased the catabolic intermediates of branched-chain amino acids, and increased SGMS1 (sphingomyelin synthase 1), which plays a role in sphingolipid metabolism.

Several studies also indicated the alteration of carbohydrate metabolism [37,78,79]. Lactate, a metabolite of carbohydrate metabolism, was shown to be increased in the plasma of trained women when compared to sedentary women [37]. Lactate produced after high cell work can act as both an energy source and a gluconeogenic precursor. With regular exercise, there is repeated lactate exposure, leading to adaptive processes such as mitochondrial biogenesis and improved urea flexibility. Therefore, the high levels of lactate in trained female basketballers indicate that the type of training in basketball induces adaptation in well-trained athletes. Another significant alteration of carbohydrate metabolism after chronic combined training is in the glycolysis pathway, evidenced in one study by the significant changes in the pyruvate levels between the training and control groups [78]. Furthermore, another study reported a decrease in the concentrations of glucose and pyruvate in the training group, indicating improved pyruvate metabolism and carbohydrate metabolism.

Amino acid metabolism was also shown to be significantly altered in several studies [37,38,78,79,81,82]. The studies summarized in this review found statistically different concentrations of amino acids involved in alanine and aspartate metabolism; glutamate metabolism; the urea cycle; arginine and proline metabolism; glycine, serine, and threonine metabolism; and histidine metabolism. One study also identified an increase in urea in plasma from athletes [2]—this, coupled with the elevations of ornithine concentration in this study and others, indicates activation of the urea cycle after chronic combined exercise [37,81,82]. Another study noted an increase in histidine levels after 24 weeks of chronic training, indicating activation of the histidine metabolism pathway and protein biosynthesis [78]. Finally, Duft et al., reported increased levels of glutamine and decreased levels of 3-hydroxyisobutyrate in the training intervention group, which indicates the involvement of glutamate metabolism and leucine metabolism, isoleucine metabolism, and valine metabolism, respectively [79]. The studies reporting the metabolic changes associated with combined chronic endurance–resistance training are summarized in Appendix A [39].

## 7. Conclusions

Exercise has numerous benefits for both physical and mental health, which are studied through metabolomics and the analysis of the metabolites released by tissues such as skeletal muscle, bone, and the liver. These metabolites play crucial roles in the energy production and respiration pathways, leading to metabolic adaptations and improved cardiovascular fitness, reduced inflammation, and increased muscle mass (Figure 1 and Figure 2). Endurance training increases mitochondrial content and oxidative enzymes, while resistance training increases muscle fiber and glycolytic enzymes. Acute endurance exercise leads to changes in amino acid metabolism, fat metabolism, and cellular energy metabolism as well as cofactor metabolism and vitamin metabolism. Subacute and chronic endurance exercise result in alterations in amino acid metabolism, lipid metabolism, and nucleotide metabolism and improved lipid metabolism. Acute resistance exercise increases succinate metabolism, lactate metabolism, pyruvate metabolism, malate metabolism, and amino acid metabolism, leading to increased anaerobic processes and muscular strength. Chronic resistance exercise affects amino acid metabolism, energy metabolism, lipid metabolism, and bone metabolism, resulting in improved cardiovascular risk factors and skeletal muscle adaptations. Combined endurance–resistance exercise leads to changes in lipid metabolism, carbohydrate metabolism, and amino acid metabolism, improved anaerobic metabolic capacity and fatigue resistance, and the involvement of purine metabolism. The study of exercise-induced metabolites is a growing field, with the potential for uncovering more metabolic mechanisms and tailoring exercise programs for optimal health and performance.

## Figures and Tables

**Figure 1 metabolites-13-00694-f001:**
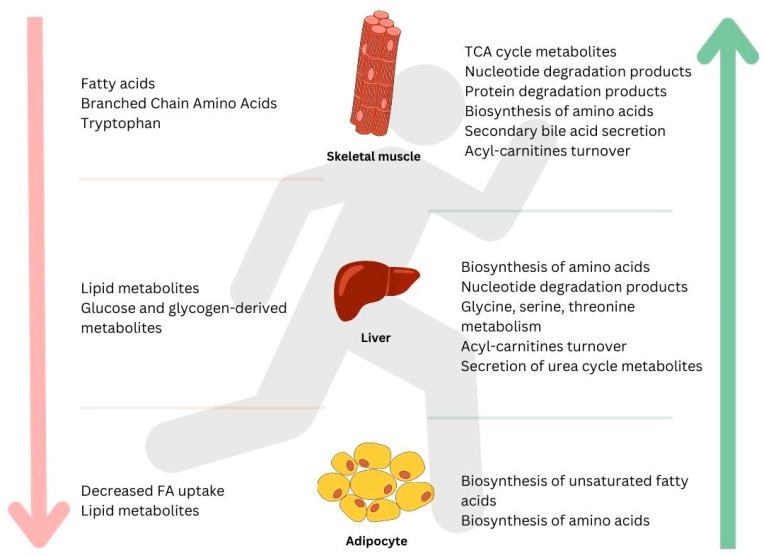
A general summary of metabolites induced by a bout of endurance exercise in different tissues: skeletal muscle, liver, adipocyte, and bone. Downward arrow on the left (↓) shows metabolites that decreased in response to exercise; upward arrow on the right (↑) indicates metabolites that increased in response to exercise.

**Figure 2 metabolites-13-00694-f002:**
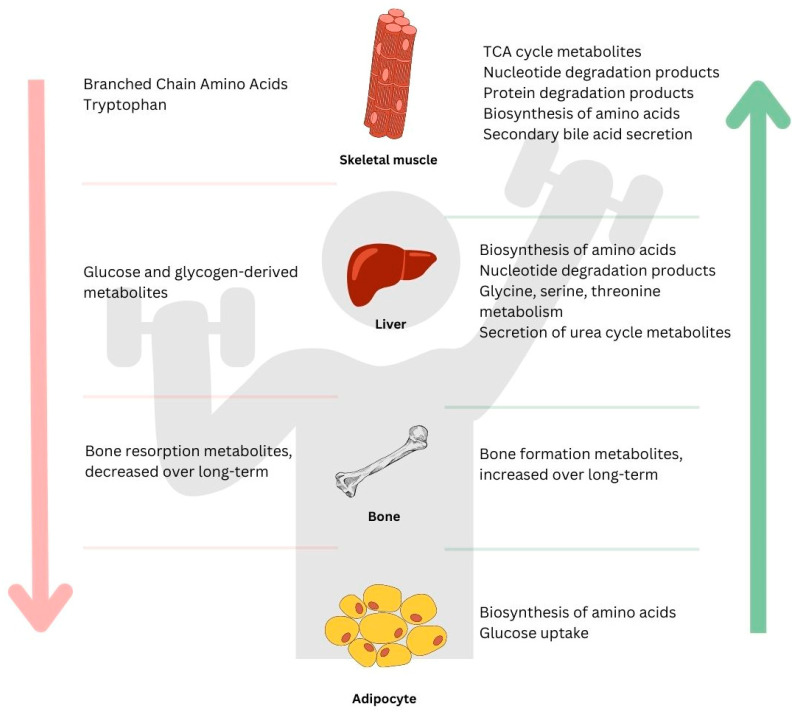
A general summary of metabolites induced by a bout of resistance exercise in different tissues: skeletal muscle, liver, adipocyte, and bone. Downward arrow on the left (↓) shows metabolites that decreased in response to exercise; upward arrow on the right (↑) indicates metabolites that increased in response to exercise.

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
