# Peer review of "Exercise Metabolome: Insights for Health and Performance"

_metabolites, 2023, doi:10.3390/metabo13060694_

Round 1

Reviewer 1 Report

The manuscript under review is in fact a review manuscript covering a vast area including medical use of metabolomics, as well descriptions of the effects of various types of exercise on metabolism. The descriptions are very vague and lack necessary details. The literature is reviewed in the tables, manuscript by manuscript. However, the descriptions make an impression of being written for personal use. E.g., what is “a reduction of metabolites associated with insulin resistance” (Koay, 2021)? Do the authors mean reduction in concentrations? Similarly, Table 3 lacks the descriptions of identified metabolic changes, despite these changes are invoked in the caption of this very table.

Even if the missing information is added, the manuscript would be too long. The authors, therefore, have a choice to make: either move the (amended) tables to supplementary material and provide this information in the text in a condensed form, or provide a table of content in the beginning of the manuscript. Nevertheless, in the latter case they should provide more details in the paragraphs describing the tables.

Other issues:

1.       Medical literature provides multiple examples that exercise has positive effects on various ailments, as well as the processes of aging. It would be helpful to provide specific examples on how metabolomics was used to unravel the underlying mechanisms. Similarly, metabolomics might be used in early diagnosis of conditions that have no treatment, such as Alzheimer’s disease. It would add to the introduction to mention these possibilities.

2.       Introduce the types of exercise and describe them in one chapter.

3.       Please add a chapter on negative effects accompanying over-exercising; add information on healthy limits of endurance exercise

4.       Language corrections necessary

5.       Interference of combined aerobic and resistance training not described

This manuscript should be reviewed by an experienced scientist.

Author Response

We would like to thank the reviewer for their very constructive and critical feedback. We have responded to your feedback below and amended the article accordingly (all changes are tracked):

Point 1: The manuscript under review is in fact a review manuscript covering a vast area including medical use of metabolomics, as well descriptions of the effects of various types of exercise on metabolism. The descriptions are very vague and lack necessary details. The literature is reviewed in the tables, manuscript by manuscript. However, the descriptions make an impression of being written for personal use. E.g., what is “a reduction of metabolites associated with insulin resistance” (Koay, 2021)? Do the authors mean reduction in concentrations? Similarly, Table 3 lacks the descriptions of identified metabolic changes, despite these changes are invoked in the caption of this very table.

Response 1:

The descriptions in all tables have been amended to reflect a greater degree of specificity in the identified metabolic changes, including for Table 3.

Koay et al. (2021) has been altered to state “reduction in the concentrations of metabolites…”.

Point 2: Even if the missing information is added, the manuscript would be too long. The authors, therefore, have a choice to make: either move the (amended) tables to supplementary material and provide this information in the text in a condensed form, or provide a table of content in the beginning of the manuscript. Nevertheless, in the latter case they should provide more details in the paragraphs describing the tables.

Response 2:

The tables have been moved to supplementary material and details of this information has been included in the text in a condensed form.

Point 3: Medical literature provides multiple examples that exercise has positive effects on various ailments, as well as the processes of aging. It would be helpful to provide specific examples on how metabolomics was used to unravel the underlying mechanisms. Similarly, metabolomics might be used in early diagnosis of conditions that have no treatment, such as Alzheimer’s disease. It would add to the introduction to mention these possibilities.

Response 1: These examples have been added to the introduction.  

Point 4: Introduce the types of exercise and describe them in one chapter.

Response 1: A chapter describing the types of exercises been added – “Types of Exercise”

Point 5: Please add a chapter on negative effects accompanying over-exercising; add information on healthy limits of endurance exercise

Response 1: A chapter describing the types of exercises been added – “Effects of Overexercising.”

Point 6: Language corrections necessary

Response 1: The language of the tables has been amended to reflect a greater degree of specificity regarding the metabolic changes.  

Point 7: Interference of combined aerobic and resistance training not described

Response 1: A paragraph on the interference of combined aerobic and resistance exercise has been included at the end of the section “Effects of Exercise Modality”.  

Reviewer 2 Report

The manuscript needs to be more focused and reduced in length by about 50%. Now, it looks more likely like a book chapter. The tables need also to be markedly reduced.

Keep also in your mind that muscle metabolic responses follow the below more or less connected criteria:

the intensity of exercise, the length of exercise, substrate availability, oxygen availability, training status, cardiovascular adaptations, age, gender, type of muscle recruited, presence of comorbidities, hydration status, diet,

Resistive exercise has hardly any cardiovascular effects.

Fig. 1 refers to what type of exercise? It does not describe all types of exercises.

Minor editing of English language required.

Author Response

We would like to thank the reviewer for their very constructive and critical feedback. We have responded to your feedback below and amended the article accordingly (all changes are tracked):

Point 1: The manuscript needs to be more focused and reduced in length by about 50%. Now, it looks more likely like a book chapter. The tables need also to be markedly reduced.

Response 1: The tables have been moved to supplementary material. The information from the tables is included in the text.

Point 2: Resistive exercise has hardly any cardiovascular effects.

Response 2:

This sentence stating that resistive exercise has cardiovascular improvement has been removed from the abstract.

Several studies in this review noted significant cardiovascular effects with resistive exercise. 

However as there is still some literature that goes against this, I decided it would be better to remove this sentence from the abstract. 

Point 3: Fig. 1 refers to what type of exercise? It does not describe all types of exercises.

Response 1: Figure 1 has been replaced with two new figures – one summarizing the general metabolic effects of endurance exercise, and the other for resistance exercise.

Reviewer 3 Report

In the manuscript entitled "Exercise Metabolome: Insights for Health and Performance" by Jaguri et al, the authors revised the up-to-date knowledge of the effect of different types (intensities) of exercise in human metabolism. The revision is well-written, and clear, compiling interesting and important aspects of the exercise.

Author Response

Thank you for your positive feedback on our manuscript. We really appreciate your recognition of the clarity and comprehensive coverage of the effects of different types and intensities of exercise on human metabolism. 

Round 2

Reviewer 1 Report

no further comments

Reviewer 2 Report

Thank you for your efforts aimed at improving the manuscript.